# Screening for antibacterial and cytotoxic activities of Sri Lankan marine sponges through microfractionation: Isolation of bromopyrrole alkaloids from *Stylissa massa*

Lakmini Kosgahakumbura[1,2☯], Jayani Gamage[1,2☯], Luke P. Robertson[2], Taj Muhammad[2], Björn Hellman[3], Ulf Göransson[2], Prabath Jayasinghe[4], Chamari Hettiarachchi[1], Paco Cárdenas[2], Sunithi Gunasekera[2]*

1 Department of Chemistry, University of Colombo, Colombo, Sri Lanka, 2 Pharmacognosy, Department of Pharmaceutical Biosciences, Biomedical Centre, Uppsala, Sweden, 3 Drug Safety and Toxicology, Department of Pharmaceutical Biosciences, Biomedical Centre, Uppsala, Sweden, 4 Marine Biological Resources Division, National Aquatic Resources Research and Development Agency (NARA), Colombo, Sri Lanka

☯ These authors contributed equally to this work.
* sunithi.gunasekera@farmbio.uu.se

## Abstract

Sri Lanka is a biodiversity hotspot and one of the richest geographical locations of marine sponges in the Indian ocean. However, the most extensive taxonomical study on Sri Lankan sponge biodiversity dates back ~100 years and only a limited number of studies have been conducted on sponge natural products. In the current study, 35 marine sponge specimens (collected from 16 sponge habitats around Sri Lanka) were identified, microfractionated and evaluated for antibacterial and anticancer assays. In total, 30 species were characterized, of which 19 species gave extracts with antibacterial and/or cytotoxic activities. Microfractionated organic extract of *Aciculites orientalis* gave the most potent antibacterial activity against *Staphylococcus aureus* and strongest lymphoma cell toxicity was exhibited by the organic extract of *Acanthella* sp. Guided by the molecular ion peaks in the bioactive fractions, large-scale extraction of *Stylissa massa* led to the isolation of three bromopyrrole alkaloids, sceptrin, hymenin and manzacidin A/C. Of these, sceptrin exhibited broad spectrum antibacterial activity against both *Escherichia coli* and *S. aureus* (MIC of 62.5 µM against both species). Based on natural product literature, seven promising species were identified as understudied. Their further exploration may lead to the discovery of structurally novel compounds.

## Introduction

To date, more than 20,000 compounds have been isolated from marine sponges (phylum Porifera) of which more than 5,300 show a diverse range of bioactivities including antimicrobial, antitumor, and anti-inflammatory activities [1]. The production of sponge bioactive compounds may be partly used by the marine sponges for chemical defences in response to various

**Data Availability Statement:** All relevant data are within the paper and its Supporting information files.

**Funding:** The research was funded by Swedish Research Council (https://www.vr.se/english.html) under an international collaborative linkage grant awarded to S. G. as the principal investigator, C.H. as the international partner and P.C. as the collaborating scientist (Award number:2017-05416). Additionally, this study made use of the NMR Uppsala infrastructure, which is funded by Department of Chemistry-BMC and the Disciplinary Domain of Medicine and Pharmacy, Uppsala University.

**Competing interests:** The authors have declared that no competing interests exist.

environmental challenges [2–4]. Sri Lanka is a hotspot of marine biodiversity [5] and one of the regions with the highest sponge biodiversity in the Indian Ocean [6–8]. However, extensive studies from the late 19[th] to early 20[th] century have been confined to few localities such as the Gulf of Mannar, Palk Bay, Galle and Trincomalee. Notably, all of these studies, including the main comprehensive study on Sri Lankan sponges by Dendy (1905) [8], focused on the taxonomy of marine sponges, long before the "explosion" of marine natural product research in the 1970s.

Bioprospecting studies indicate that marine sponges are one of the most prolific marine sources of anticancer compounds [9]. For example, some marine sponge metabolites and/or their synthetic derivatives, including eribulin mesylate [10], panobinostat [10], cytarabine [10] and gemcitabine [11] have progressed from the laboratory to market as FDA approved anticancer drugs. Similarly, marine sponges are well-known as producers of antibacterial specialized metabolites (examples: dibromoageliferin, ageliferin A and B) with some exhibiting potent activity against multi-drug resistant bacterial strains [12]. For these reasons, this study focused on antibacterial and anticancer activities to find new bioactive specialized metabolites from Sri Lankan marine sponges. Over the past 40 years, natural products from Sri Lankan marine sponges have only been characterized in a handful of studies [13–17]. Difficulties in the accessibility of sponge habitats and lack of taxonomic expertise have led to Sri Lankan marine sponge fauna being largely understudied. Thus, the current work aimed to revisit this sponge diversity and assess its pharmacological potential in order to open new avenues for drug development in Sri Lanka.

To investigate the antibacterial activities of Sri Lankan marine sponges, an agar disc diffusion assay on the crude sponge extracts was compared with a more sensitive microdilution assay on microfractionated sponge extracts. Additionally, the anticancer activity of the microfractionated extracts was assessed in a fluorometric microculture cytotoxicity assay (FMCA). Biologically active fractions were then analyzed by LC-MS to identify molecular ion peaks potentially associated with bioactive compounds. These data allowed the isolation of three known bromopyrrole alkaloids (sceptrin, hymenin and manzacidin A or C) from *Stylissa massa* (Carter, 1887), of which sceptrin showed moderate antibacterial activity. Additionally, a list of Sri Lankan sponge species that warrant further investigation was compiled based on their antibacterial and cytotoxic activities at the crude extract level.

## Material and methods

### Field sampling and identification

Written consent to collect sponge samples from the East and Northern areas of Sri Lanka was obtained from the Department of Wildlife Conservation by the National Aquatic Resources and Research and Development Agency (NARA), Sri Lanka. Sixteen sponge samples were collected in shallow waters by scuba-diving or snorkelling in Trincomalee, Mannar and Jaffna (Table 1, S1 Fig in S1 File). Specimens were photographed and fixed directly in the field in 98% ethanol and stored at the Department of Chemistry, University of Colombo. Another 19 samples were collected as by-catch of experimental bottom trawling of Research Vessel *Dr. Fridtjof Nansen* ecosystem survey in coastal waters in Sri Lanka, during June-July 2018, at depths between 21 and 80 meters (Table 1, S1 Fig in S1 File). These specimens were frozen on board and stored in the freezer at the National Aquatic Resources Research and Development Agency (NARA) in Colombo for further processing.

Marine sponge samples were identified by looking at their siliceous or spongin skeleton under a light microscope. A small piece of sponge (2x2 mm) was digested in bleach for 30 min to 2 hrs. The remaining spicules/spongin were then washed with water and ethanol and

**Table 1. Details of the specimens investigated: Species identification, location (locality name and coordinates), depth of collection (in meters), voucher museum numbers and GenBank accession numbers.**

| No | Sample label | Species | Location | Depth | Voucher No | | CO1/and 28S |
|----|----|----|----|----|----|----|----|
| | | | | | Museum Reg No | Division Reg No | |
| 1 | 290618-12-01 (7) | *Rhabderemia indica* | Kuchchaveli 8.8635 N 81.1462 E | 41 | 2022.01.01.NH | NMSL MS IPS 001 | |
| 2 | 010718-18-01 (S02) | *Rhabdastrella globostelletta* | Batticaloa 7.8063 N 81.6943 E | 40 | 2022.01.02.NH | NMSL MS IPS 002 | OQ943998 |
| 3 | 020718-24-04 (15) | *Petrosia (Petrosia)* sp. 1 | Nintavur 7.3833 N 81.9697 E | 52 | 2022.01.03.NH | NMSL MS IPS 003 | |
| 4 | 020718-26-02 (17) | *Petrosia (Petrosia)* sp. 2 | Akkaraipattu 7.1995 N 81.9923 E | 55 | 2022.01.04.NH | NMSL MS IPS 004 | |
| 5 | 020718-26-05 (19) | *Erylus* sp. | Akkaraipattu 7.1995 N 81.9923 E | 55 | 2022.01.05.NH | NMSL MS IPS 005 | OQ943823 |
| 6 | 070718-29-05 (24) | *Aulospongus gardineri* | Galle 5.9207 N 80.1795 E | 78 | 2022.01.06.NH | NMSL MS IPS 006 | OQ944001 |
| 7 | 070718-29-08 (27) | *Manihinea* sp. | Galle 5.9207 N 80.1795 E | 78 | 2022.01.07.NH | NMSL MS IPS 007 | OQ943997, OQ943824 |
| 8 | 070718-29-11 (30) | *Halichondria* sp. | Galle 5.9207 N 80.1795 E | 78 | 2022.01.08.NH | NMSL MS IPS 008 | |
| 9 | 090718-31-02 (37) | *Spongosorites* sp. | Wadduwa 6.616 N 79.7508 E | 50 | 2022.01.09.NH | NMSL MS IPS 009 | |
| 10 | 090718-31-04 (39) | *Acanthella* sp. | Wadduwa 6.616 N 79.7508 E | 50 | 2022.01.10.NH | NMSL MS IPS 010 | |
| 11 | 100718-32-01 (S01) | *Axinella donnani* 1 | Wennappuwa 7.354 N 79.6388 E | 26 | 2022.01.11.NH | NMSL MS IPS 011 | OQ943825 |
| 12 | 070718-29-03 (22) | *Axinella donnani* 2 | Galle 5.9207 N 80.1795 E | 78 | 2022.01.12.NH | NMSL MS IPS 012 | OQ944000, OQ943826 |
| 13 | 020718-26-06 (S03) | *Topsentia* sp. | Akkaraipattu 7.1995 N 81.9923 E | 55 | 2022.01.13.NH | NMSL MS IPS 013 | OQ944003 |
| 14 | 270618-10-01 (2) | *Callyspongia (Cladochalina)* sp. | Point Pedro—Palk Strait 10.0548 N 80.5442 E | 80 | 2022.01.14.NH | NMSL MS IPS 014 | |
| 15 | 290618-11-01 (8) | *Amorphinopsis foetida* | Pulmoddai 9.0025 N 81.0023 E | 21 | 2022.01.15.NH | NMSL MS IPS 015 | OQ943999 |
| 16 | 020718-24-02 (13) | *Aciculites orientalis* | Nintavur 7.3833 N 81.9697 E | 52 | 2022.01.16.NH | NMSL MS IPS 016 | |
| 17 | 020718-24-03 (14) | *Agelas ceylonica* | Nintavur 7.3833 N 81.9697 E | 52 | 2022.01.17.NH | NMSL MS IPS 017 | |
| 18 | 070718-29-10 (29) | *Amorphinopsis* sp. | Galle 5.9207 N 80.1795 E | 78 | 2022.01.18.NH | NMSL MS IPS 018 | |
| 19 | 080818-04-06 (E41) | *Stylissa massa* 1 | Hospital beach Trincomalee 8.5652 N 81.255 E | 10–11, 3–7 | 2022.01.19.NH | NMSL MS IPS 019 | |
| 20 | 200220-01-11 | *Stylissa massa* 2 | Silavathurai, Mannar (Achchankulam) 8.8007 N 79.8498 E | 1.5–5 | 2022.01.20.NH | NMSL MS IPS 020 | |
| 21 | 070818-04-02 (E18) | *Plakinastrella ceylonica* | Hospital beach Trincomalee 8.5652 N 81.255 E | 10–11, 3–7 | 2022.01.21.NH | NMSL MS IPS 021 | OQ943827 |
| 22 | 050718-27-01 (67) | *Xestospongia testundinaria* 1 | Hambantota 5.9225 N 81.1907 E | 68 | 2022.01.22.NH | NMSL MS IPS 022 | |
| 23 | 050718-27-02 (65) | *Xestospongia testundinaria* 2 | Hambantota 5.9225 N 81.1907 E | 68 | 2022.01.23.NH | NMSL MS IPS 023 | OQ944002, OQ943828 |
| 24 | 200220-01-01 | *Haliclona (Reniera)* sp.1 | Silavathurai, Mannar (Achchankulam) 8.8007 N 79.8498 E | 1.5–5 | 2022.01.24.NH | NMSL MS IPS 024 | |
| 25 | 200220-01-02 | *Paratetilla bacca* | Silavathurai, Mannar (Achchankulam) 8.8007 N 79.8498 E | 1.5–5 | 2022.01.25.NH | NMSL MS IPS 025 | OQ943996 |

*(Continued)*

**Table 1.** (Continued)

| No | Sample label | Species | Location | Depth | Voucher No | | CO1/and 28S |
|----|-----------|---------|----------|-------|-----------|-----------|-----------|
| | | | | | Museum Reg No | Division Reg No | |
| 26 | 200220-01-04a | *Stellitethya* sp. | Silavathurai, Mannar (Achchankulam) 8.8007 N 79.8498 E | 1.5–5 | 2022.01.26.NH | NMSL MS IPS 026 | |
| 27 | 200220-01-10 | *Ircinia* sp. | Silavathurai, Mannar (Achchankulam) 8.8007 N 79.8498 E | 1.5–5 | 2022.01.27.NH | NMSL MS IPS 027 | |
| 28 | 210220-02-07 | *Cliona* sp. | South Bar, Mannar 8.9240 N 79.7859 E | 2–2.5 | 2022.01.28.NH | NMSL MS IPS 028 | OQ943829 |
| 29 | 220220-03-02 | *Haliclona* sp. | Chaddy beach, Jaffna 9.6265 N 79.9248 E | 1.5 | 2022.01.29.NH | NMSL MS IPS 029 | |
| 30 | 220220-03-04 | *Spheciospongia inconstans* 1 | Chaddy beach, Jaffna 9.6265 N 79.9248 E | 1.5 | 2022.01.30.NH | NMSL MS IPS 030 | OQ943830 |
| 31 | 220220-03-05 | *Haliclona (Reniera)* sp.2 | Chaddy beach, Jaffna 9.6265 N 79.9248 E | 1.5 | 2022.01.31.NH | NMSL MS IPS 031 | OQ943831 |
| 32 | 220220-03-08 | *Spheciospongia inconstans* 2 | Chaddy beach, Jaffna 9.6265 N 79.9248 E | 1.5 | 2022.01.32.NH | NMSL MS IPS 032 | |
| 33 | 230220-04-05 | *Haliclona (Reniera)* sp.3 | Allaipiddy, Velanai Island, Jaffna 9.6117 N 79.9584 E | < 1 | 2022.01.33.NH | NMSL MS IPS 033 | OQ943832 |
| 34 | 230220-04-07 | *Spheciospongia inconstans* 3 | Allaipiddy, Velanai Island, Jaffna 9.6117 N 79.9584 E | <1 | 2022.01.34.NH | NMSL MS IPS 034 | |
| 35 | 230220-05-06 | *Tedania (Tedania)* sp. | Kayts Island, Jaffna 9.7025 N 79.8639 E | 0–1 | 2022.01.35.NH | NMSL MS IPS 035 | |

embedded in Eukitt® mounting medium (Sigma-Aldrich, USA) on a glass slide. Identification keys from the *Systema Porifera* [18] were used to identify family and genus; the latest demosponge classification from the World Porifera Database (WPD) (https://www.marinespecies.org/porifera) was followed. By using the taxonomy literature available in the WPD, some of the specimens were identified to the species level. Spicule and/or fibre preparations of specimens (i.e., voucher slides) are deposited at the Zoology division of the Colombo National Museum, Sri Lanka (see Table 1 for museum reference numbers).

The DNA of the preserved samples were extracted using the DNeasy® Blood and Tissue kit (Qiagen). A 658 base pair region of the 5' end of CO1 (Folmer fragment) and ~800 base pair region of 28S rRNA (the C1-D2 region) genes were amplified using Polymerase Chain Reaction (PCR) following the protocol in Cárdenas et al. (2011) [19]; sequencing was done at Macrogen Inc. (Beotkkot-ro, Geumcheon-gu, Seoul, Korea). Sequences were blasted using the blastn tool with the NCBI database to confirm our morphological identifications or narrow down more challenging identifications to the lowest possible taxonomic level.

### Preparation of crude extracts

Sponge specimens were washed thoroughly to remove debris from sea water and then freeze dried. The ground, dried sponge material from each specimen (~30 g) was sequentially extracted with 250 ml of solvent in the following order: (i) aqueous (60% MeOH), (ii) organic (DCM:MeOH (v/v 9:1)) and (iii) ethanolic (96% EtOH). The crude extract was obtained after filtering and solvent evaporation using a rotary evaporator. After each solvent extraction, the residual sponge material was air dried before proceeding to the next solvent of extraction. Extraction was repeated twice for every specimen.

## Agar disc diffusion assay

An established protocol for an agar disc diffusion assay was followed [20]. The bioactivities of crude extracts were tested against two Gram-positive (*Staphylococcus aureus* (ATCC 25928), and *Bacillus cereus* (ATCC 11778)) and two Gram-negative bacteria (*Pseudomonas aeruginosa* (ATCC 9027), and *Escherichia coli* (ATCC 35218)). A cell suspension was prepared for each bacterial culture in sterile saline (0.9% NaCl, Sigma-Aldrich, Korea). After adjusting the turbidity, each cell suspension (200 μl) was re-distributed evenly on solidified Muller Hinton Agar plates (HiMedia Laboratories, India; 38.0 g in 1000 ml of distilled water). The aqueous and ethanolic extracts were dissolved in distilled water while the organic crude extracts were dissolved in 10% DMSO to a final concentration of 1000 μg/ml. Twenty microlitres of each was then transferred to a sterile disc (Whatman grade A filter paper discs of 6 mm diameter). The solvent without crude extract was used as the negative control and ampicillin (10 μg/disc) and gentamicin (10 μg/disc) were used as positive controls. The diameter of the inhibition zone was then measured after incubation overnight at 37 ºC. The entire assay was carried out in triplicate.

## Microfractionation and preparation of microtiter plates

Two milligrams from the aqueous and ethanolic extracts were fractionated into 48 fractions (1 ml x 48) in a 96 position deep well plate (VWR, Sweden) using a Shimadzu LC-10 system equipped with an SPD-M10AVP Photodiode-Array (PDA) detector. A Phenomenex Jupiter C18 HPLC column (5 μm, 100 Å, 4.6 mm × 250.0 mm) was used for the microfractionation of the crude extracts by employing a gradient of 5–95% $CH_3CN$ (0.05% TFA) over 48 minutes at a flow rate of 1 ml/min. Similarly, 10 mg of the organic extracts were dissolved in 1 ml of 100% $CH_3CN$ and a volume of 200 μl was fractionated into 48 fractions using two consecutive gradients from 5–60% $CH_3CN$ (0.05% TFA) over 20 minutes and 60–95% $CH_3CN$ (0.05% TFA) over 25 minutes at a flow rate of 1 ml/min [20]. One hundred microlitres from each well was transferred into 96-well U-bottom polystyrene plates (Greiner Bio-one, USA) and dried in a centrifugal evaporator (Savant Speed Vac plus SC110A).

## Microdilution assay

The dried 96 position U-bottom plates prepared after microfractionation of aqueous, ethanolic and organic extracts were tested for antibacterial activity against *S. aureus* (ATCC 29213) and *E. coli* (ATCC 25922) in duplicate. Bacterial strains were obtained from the Department of Clinical Bacteriology at Lund University Hospital. An established protocol was followed [21]. Microbial strains were first cultured at 37 ºC to mid-logarithmic phase on 3% tryptic soy broth (TSB, Merck KGaA, Germany). Bacterial cells were washed twice by centrifugation and re-suspension in tris buffer (10 mM, pH 7.8 at RT) to 100,000 CFU/ml (measured at OD600). To the wells containing dried microfractionated aqueous and ethanolic extracts, 50 μl of tris buffer and 50 μl of bacterial suspension was added. The wells containing dried microfractionated organic extracts were first dissolved in 5 μl of DMSO and then 45 μl of tris buffer followed by addition of 50 μl of the bacterial suspension (final DMSO concentration per well was 5%). After 5 hrs of incubation at 37 ºC, each well was administered 5 μl of 20% TSB (low temp sterilised at 100 ºC) and the plates were re-incubated for 6–9 hrs depending on the growth rate of each bacterial strain used. A volume of 50 μl of tris buffer and 50 μl of the bacterial suspension were used as the negative control and the human antibacterial peptide LL-37 (5 μM) was used as the positive control. The potency of the antibacterial activity was recorded by visible inspection of the wells exhibiting a total growth inhibition. Plates that showed bacterial growth

inhibition were incubated for further six hours to ensure that growth inhibition was maintained (up to 12 hrs for *E. coli* and 15 hrs for *S. aureus*).

Once the most antibacterial extracts were identified, microfractionation was repeated for the extracts available in sufficient quantities and the antibacterial activity of the fractions was tested in a two-fold serial dilution. A volume of 200 μl from each bioactive fraction was transferred to 96 well plates and the plates were dried using the centrifugal evaporator. One hundred microlitres of tris buffer was added to each dried bioactive fraction, and the rest of the wells were filled with 50 μl of tris buffer. A two-step serial dilution was carried out followed by the addition of 50 μl of bacterial suspension to each well. Introduction of 20% TSB and incubation was carried out as described above.

## Muller hinton broth (MHB) microdilution assay for purified compounds

The pure compounds isolated from *Stylissa massa* (specimen 200220-01-11, Silavathurai Reef, Gulf of Mannar) were subjected to a standard MHB microdilution assay [21] against *S. aureus* (ATCC 29213) and *E. coli* (ATCC 25922). The growth media was introduced initially by the addition of 50 μl of MHB (Merck KGaA, Darmstadt, Germany) (2.1% (w/v)) to each well. The two-step serial dilution was carried out for pure compounds sceptrin and hymenin as described above with a starting concentration of 250 μM in a final volume of 100 μl followed by the addition of 50 μl of bacterial suspension. One hundred microlitres of media was used as the negative control and ciprofloxacin (5 μM) was used as the positive control. Plates were incubated at 37 ºC and interpretation was done after 16–20 hrs. The minimum inhibitory concentration (MIC) value was defined as the lowest concentration showing total growth inhibition upon visual inspection.

## Fluorometric microculture cytotoxicity assay (FMCA)

Nineteen crude extracts, subfractions prepared from the aforementioned crude extracts and two pure compounds (sceptrin and hymenin) were evaluated for cell toxicity using FMCA as described previously [20]. Cytotoxicity was performed using the human lymphoma cell line U-937 GTB. The cells were grown in RPMI-1640 (Sigma-Aldrich, USA) supplemented with 10% heat-inactivated fetal calf serum (Sigma-Aldrich, USA), 2mM glutamine, 50 μg/ml streptomycin and 60 μg/ml penicillin. Cells were harvested during the exponential growth phase. Crude extracts were screened at final concentrations between 100–400 μg/ml and pure compounds between 12.5–100 μM.

For pure compounds and extracts, 20 μl of each was added to a V-bottom plate and the wells seeded with 180 μl of cell suspension. V-bottom plates prepared after microfractionation were seeded with 200 μl of cell suspension containing approximately 20,000 cells per well. The plates were incubated at 37 ºC in a 5% $CO_2$ atmosphere for 72 hrs. Centrifugation of the plates, washing of the cells and incubation for 40 minutes at 37 ºC after addition of fluorescein diacetate was carried out as described previously [20]. The fluorescence of each well was measured (Thermoscientific Varioskan Flash) with an excitation and emission of 485 and 538 nm, respectively. The fluorescence of each well is proportional to the number of living cells and thus the cytotoxic activity is inversely proportional to the fluorescence. The activities of the extracts are reported in terms of percentage Survival Index (%SI).

$$\%\text{SI} = \left\{ \frac{(\text{Fluoresence in the exp. wells} - \text{Avg fluoresence in the blank wells})}{(\text{Avg fluoresence in the control wells} - \text{Avg fluoresence in the blank wells})} \right\} x100 \quad (1)$$

## Fractionation and isolation of specialized metabolites from *Stylissa massa* ethanolic extract

Approximately eight grams of the ethanolic extract of *Stylissa massa* (voucher numbers: Museum Reg No- 2022.01.20.NH Division Reg No- NMSL MS IPS 019) was further fractionated using a ÄKTA FPLC (Amersham Pharmacia Biotec) fitted to a Biotage $C_{18}$ spherical flash column (100 Å, 40 g, 40–60 μm) with UV detection at 254 nm using a gradient from 100% $H_2O$ (0.1% TFA) to 100% $CH_3CN$ (0.1% TFA) over 60 mins at a flow rate of 10 ml/min. Fraction collection was performed every 60 seconds. Fractions 13–14, 15–18 and 19–29 were separately recombined. Further purification was carried out using semi preparative RP-HPLC with a Phenomenex Jupiter $C_{18}$ column (5 μm, 300Å, 10.0 mm x 250.0 mm) at a flow rate of 3 ml/min for 55 minutes using a linear gradient from 5–50% $CH_3CN$ (0.05% TFA). Fractions were analysed by (+)-LRESI-MS using a Thermo Finnigan LCQ, and fractions containing similar molecular ion peaks were combined. Analytical RP-HPLC to determine the purity of fractions was carried on a Phenomenex Kinetex XB-$C_{18}$ column (5 μm, 100 Å, 4.6 mm × 250.0 mm) using a linear gradient from 5–97% $CH_3CN$ (0.05% TFA). The accurate mass of each pure compound was analysed using a Xevo G2-XS quadrupole time-of-flight (QToF) mass spectrometer coupled with a nano-Acquity UPLC (Waters Corp. Milford, MA, USA). Structure characterization of the pure compounds was carried out using NMR data acquired on a Bruker Avance Neo 600 MHz (TCI (CRPHe TR-1H and 19F/13C/15N 5 mm-EZ)) spectrometer at 298 K. All compounds were dissolved in methanol-$d_4$.

## Results

### Identification of Sri Lankan sponge samples

A total of 35 samples (Table 1), one Homosclerophorida and 34 Demospongiae (S1 Fig in S1 File), were collected from 16 locations around Sri Lanka (S2 Fig in S1 File). The 34 demosponges belonged to 12 orders: Tetractinellida, Biemnida, Axinellida, Bubarida, Suberitida, Agelasida, Scopalinida, Clionaida, Haplosclerida, Dictyoceratida, Poecilosclerida and Tethyida. Of the 35 specimens, 17 were identified up to species level and the rest were identified up to the genus level, combining morphological data (external shape, spicules and/or spongin) and molecular markers (Folmer fragment of CO1 and/or 28S rRNA genes) (Table 1). Most of the species were reported from Sri Lanka before [8, 22, 23] but five species are not recorded previously from Sri Lanka (Table 1). At least two specimens may represent new species (*Erylus* sp. and *Manihinea* sp.), which will be described in a separate publication.

### Antibacterial activity of crude sponge extracts

To identify the antibacterial activity of extracts, all the crude extracts were first subject to an agar disc diffusion assay against *S. aureus*, *B. cereus*, *P. aeruginosa* and *E. coli* (S3 Fig in S1 File). In the disc diffusion assay, the minimum concentration which showed the antibacterial activity for sponge extract is 20 μg/disc. Hence, all sponge extracts at concentrations above 20 μg/disc were screened for antibacterial activity. Of 51 extracts tested, seven were identified to have moderate antibacterial activity, with inhibition zone diameters between 7 ± 0.3 and 9 ± 0.3 mm at 20 μg per disc (Table 2). Notably, out of these extracts, six demonstrated activity against *E. coli*, whereas the remaining extract exhibited activity against *S. aureus*. The positive controls, ampicillin (10 μg disc) and gentamicin (10 μg disc) showed an inhibition zone diameter around 20 mm ± 0.3 against *S. aureus* and *E. coli* respectively. None of the extracts exhibited activity against *B. cereus* and *P. aeruginosa*.

**Table 2. Summary of the antibacterial activity of sponge specimens subjected to disc diffusion and micro dilution assays.**

| No | Specimen | Type of the extract | Disc diffusion assay | | Microdilution assay | |
|---|---|---|---|---|---|---|
| | | | Inhibited bacterial strain | Mean diameter of the inhibition zone (mm) | Bioactive fractions against *S. aureus* (major mass to charge ratios (*m/z*) observed in the bioactive fractions) | Bioactive fractions against *E. coli* (major mass to charge ratios (*m/z*) observed in the bioactive fractions) |
| 1 | *Rhabderemia indica* | organic | X[#] | | 34–36, (1043.44, 885.58) 42–46 (1792.69, 885.83) | X |
| 2 | *Rhabdastrella globostelletta* | organic | *E. coli* | 9 ± 0.3 | X | X |
| 3 | *Petrosia (Petrosia)* sp. 1 | organic | X | | 18–19 (1039.77) 29–46 (353.44) | X |
| 4 | *Petrosia (Petrosia)* sp. 2 | aqueous | X | | X | 8 No identified prominent mass/masses |
| | | organic | X | | 37–39 42–46 (1791.75 might be the sodium ion of dimerized 885.98, 885.98) | X |
| 5 | *Erylus* sp. | aqueous | X | | *34–39 (1065.68, 1095.69) | X |
| 6 | *Aulospongus gardineri* | aqueous | X | | 12–13, 39 (735.15) slightly | X |
| | | organic | X | | 33–39 (885.35, 1791.36) | X |
| 7 | *Halichondria* sp. | aqueous | X | | X | 20 (1387.75, 1039.84) |
| | | organic | X | | *23–24 (1086.59, 328.59) | X |
| 8 | *Acanthella* sp. | aqueous | *S. aureus* | 8 ± 0.3 | 26–40, 44–45 (848.78, 415.55) | X |
| 9 | *Axinella donnani* 1 | aqueous | *E. coli* | 8 ± 0.3 | X | X |
| | | organic | *E. coli* | 7 ± 0.3 | X | X |
| 10 | *Topsentia* sp. | organic | X | | *29–44 (497.86) | X |
| 11 | *Callyspongia (Cladochalina)* sp. | organic | X | | *41–43, (421.42, 885.81) 46 (281.55, 309.63) | X |
| 12 | *Aciculites orientalis* | organic | X | | *9,31 (222.20, 295.52, 595.79, 355.39) | 10–12, 17-21(222.20) |
| 13 | *Agelas ceylonica* | organic | X | | 30–32 slightly (401.85, 470.92) | X |
| 14 | *Amorphinopsis* sp. | organic | X | | 26–46 (1931.70, 393.61, 1792.64) | X |
| 15 | *Stylissa massa* 1 | ethanolic | *E. coli* | 7 ± 0.3 | 1–2, (324.23) 15–19 (621.34, 1037.80) | 2 (324.23) 16–18 (621.34, 1037.80) |
| 16 | *Xestospongia testundinaria* 1 | aqueous | *E. coli* | 8 ± 0.3 | X | 7 (421.40) |
| | | organic | *E. coli* | 10 ± 0.3 | 36–37 No identified prominent mass/masses | X |
| 17 | *Spheciospongia inconstans* 1 | aqueous | X | | 3–4 (202.09, 389.94, 242.98, 446.21) 32 | 7–8 (204.09) 14,15 (979.64) 16 (4913.20, 4897.24) 26–27 (324.01) |
| 18 | *Spheciospongia inconstans* 2 | aqueous | X | | 5–6 (230.24, 385.13) 32 No identified prominent mass/masses | X |

*(Continued)*

**Table 2.** (Continued)

| No | Specimen | Type of the extract | Disc diffusion assay | | Microdilution assay | |
|---|---|---|---|---|---|---|
| | | | **Inhibited bacterial strain** | **Mean diameter of the inhibition zone (mm)** | **Bioactive fractions against *S. aureus* (major mass to charge ratios (*m/z*) observed in the bioactive fractions)** | **Bioactive fractions against *E. coli* (major mass to charge ratios (*m/z*) observed in the bioactive fractions)** |
| 19 | *Tedania (Tedania)* sp. | ethanolic | X | | 38 (137.08) | X |

*The bioactive fractions selected for two-step microdilution assay

#X denotes extracts that were screened but gave no antibacterial activity

Two milligrams of each crude extract were microfractionated into 48 fractions using RP-HPLC and the antibacterial activity of the fractions were assessed by a microdilution assay (Fig 1) (Table 2). The antibacterial fractions were further analyzed by LC-MS to identify the molecular ion peaks associated with bioactive compounds (Table 1). Twenty-one extracts (derived from a total of 16 species) were antibacterial in the microdilution assay, despite only seven extracts originating from four species inhibiting bacteria in the disc diffusion assay. The assay results of the 27 extracts with no antibacterial activity are compiled in S1 Table in S1 File. Importantly, the aqueous extract of *Erylus* sp. and organic extracts of *Callyspongia (Cladochalina)* sp., *Topsentia* sp., *Petrosia (Petrosia)* sp. 1, *Petrosia (Petrosia)* sp. 2, *Amorphinopsis* sp. and *Acanthella* sp., were the most potent antibacterial extracts (no visible growth against *S. aureus* up to 15 hrs when incubated at 37 ºC) identified from the microdilution assay.

The most potent antibacterial extracts available in sufficient quantities were further tested in a two-step serial dilution assay. Here, the organic extract of *A. orientalis* (Dendy, 1905) was highlighted as the most potent extract with a significant antibacterial activity against *S. aureus* (inhibitory activity observed down to 32-fold dilution of the fractions). Similarly, microfractionated organic extracts of *Callyspongia (Cladochalina)* sp., *Topsentia* sp. and *Halichondria* sp. inhibited *S. aureus* down to a 16-fold diluted concentration. All of the above active extracts were available in inadequate amounts (<500 mg), limiting any follow-up compound isolation.

Of the extracts that were available in larger amounts (~8 g), *Stylissa massa* showed significant antibacterial activity for its ethanolic extract. A total growth inhibition was observed for the bioactive fractions 1–2 and 15–19 in the microdilution assay. The monoisotopic masses *m/z* 324.2352, 621.3403 and 1037.8091 were detected respectively in the bioactive fractions 2, 15 and 19 (S4 Fig in S1 File). Therefore, *Stylissa massa* was selected for large scale isolation and characterization of bioactive natural products.

## Lymphoma cell toxicity assay of the crude sponge extracts

The fluorometric microculture cytotoxicity assay (FMCA) is a nonclonogenic microplate-based cell viability assay used for measurement of the cytotoxicity of compounds *in vitro*. Of the screened 19 crude extracts, seven were identified to be cytotoxic (average %SI below ~ 30% at 400 μg/ml). These were the extracts from *Xestospongia testudinaria* (Lamarck, 1815), *Erylus* sp., *Axinella donnani* (Bowerbank, 1873), *Manihinea* sp., *Acanthella* sp. and *Stylissa massa* (Table 3). These seven crude extracts were then microfractionated to identify bioactive fractions and the dominant molecular ion peaks with MS. The resulting bioactive fraction numbers 30–31 and 39 of the organic extract of *Acanthella* sp. (with an average %SI below ~ 30%

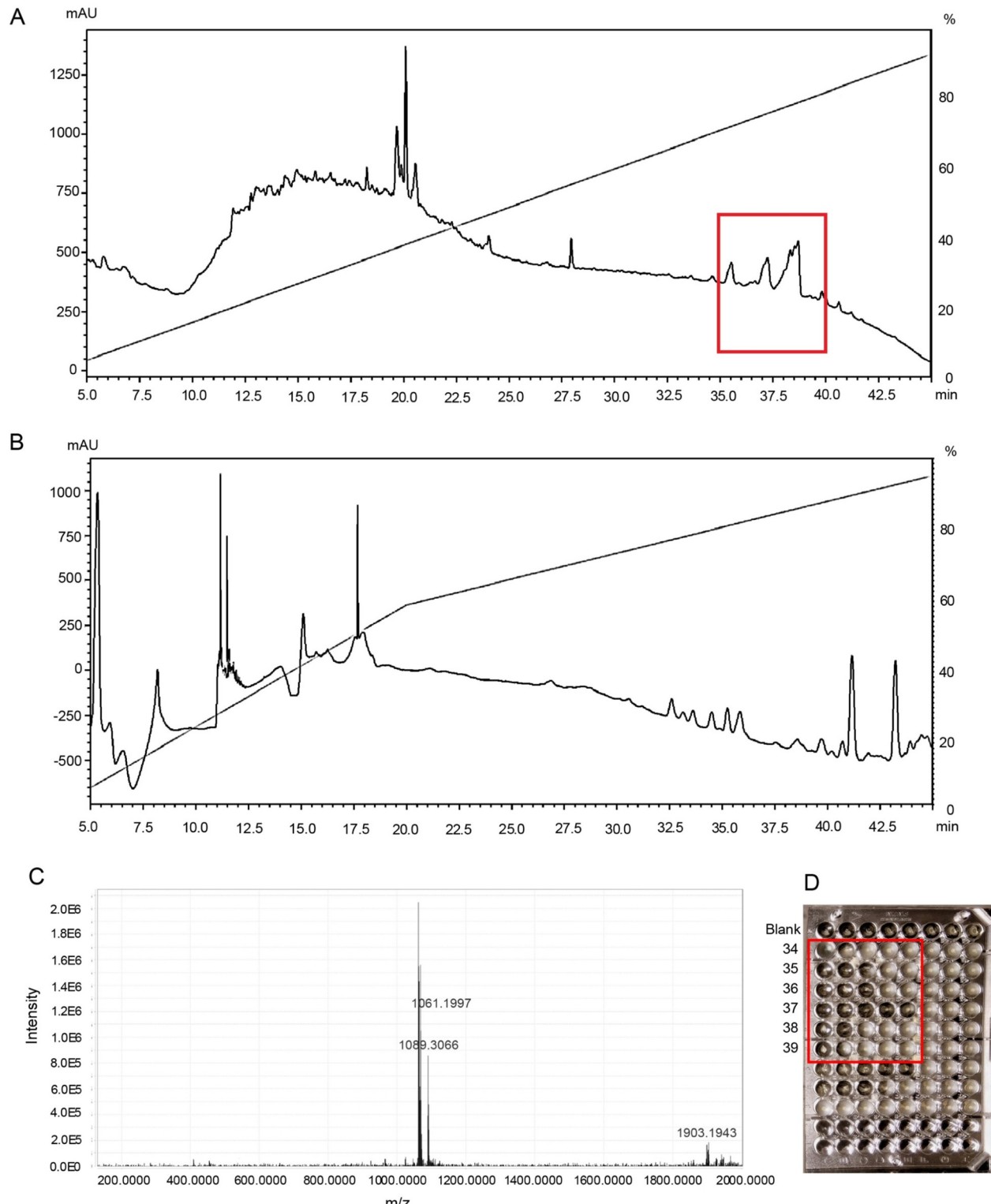

**Fig 1. The bioassay guided microfractionation protocol followed for the identification of molecular ions in sponge extracts, exemplified by** ***Erylus* sp.** A: The RP-HPLC chromatogram of microfractionated aqueous extract of *Erylus* sp. (254 nm) with the antibacterial region highlighted in a red square. B: The RP-HPLC chromatogram of the microfractionated organic extract of *Erylus* sp. (254 nm). C: The dominant molecular ions in bioactive fraction 37 of *Erylus* sp. D: The microdilution assay against *S. aureus* for *Erylus* sp. showing bacterial inhibition down to a two-fold dilution for the fractions 34 and 38 (inhibition in the first two wells from left), down to a four-fold dilution for the fractions 35 and 36 (inhibition in the first

three wells from left), down to a sixteen-fold dilution for the fraction 37 (inhibition down to five wells from left) and fraction 39 with no dilution (inhibition only in the first well from the left).

and ~ 6% respectively) showed the strongest cytotoxicity. The other extracts showed no cyto-toxicity after microfractionation (% SI between 50 to 65%).

## Large-scale isolation, structure characterization and bioactivity screening of compounds from *Stylissa massa*

Since the ethanolic extract of *S. massa* showed prominent antibacterial and cytotoxic activities, it was selected for large-scale extraction. A standard compound isolation protocol was used as shown in Fig 2.

Following the RP-MPLC fractionation, only one monoisotopic mass *m/z* 619.1271 was successfully purified from fractions 20–30 which was observed from the bioactive wells during microfractionation process. Unfortunately, *m/z* 324.2352 and *m/z* 1037.8091 could not be isolated as they were present in insufficient quantities. The ionization pattern of the purified fractions in ESI-MS revealed the presence of brominated alkaloids as exemplified in S4 Fig in S1 File. Analysis of these fractions by 2D NMR and (+)-HRESI-MS led to the identification of

**Table 3. Results of FMCA against the histiocytic human lymphoma cell line U-937.**

| Species name[1] | Type of the extract | Concentration/µg/ml | %SI of the crude extract | Bioactive fractions and the major mass to charge ratios (*m/z*)[2] | %SI of the bioactive fractions[3] |
|---|---|---|---|---|---|
| *Xestospongia testudinaria* 2 | aqueous | 400 | 17.96 | 32 (376.28) | 65.75 |
| | | 200 | 34.72 | | |
| | | 100 | 88.75 | | |
| *Erylus* sp. | organic | 400 | 32.13 | 18 (712.42) | 66.62 |
| | | 200 | 74.24 | | |
| | | 100 | 138.68 | | |
| *Axinella donnani* 2 | organic | 400 | 16.43 | 33 (376.26, 200.23, 230.24, 561.49) | 54.08 |
| | | 200 | 99.04 | | |
| | | 100 | 109.58 | | |
| *Manihinea* sp. | organic | 400 | 0.01 | 3–5 (296.03, 304.03) | 59.16 |
| | | 200 | 7.62 | | |
| | | 100 | 24.08 | | |
| *Acanthella* sp. | organic | 400 | 0.46 | 30–31 (373.31, 269.25) | 29.45 |
| | | 200 | 3.51 | | |
| | | 100 | 59.96 | 39 (415.2, 356.27) | 6.04 |
| *Stylissa massa* 2 | aqueous | 400 | 9.76 | 4,6 (324.00) | 56.04 |
| | | 200 | 12.66 | | |
| | | 100 | 37.69 | 27–30 (202.08) | 53.23 |
| *Stylissa massa* 2 | ethanolic | 0.4 | 1.56 | 13,14 (799.46, 374.24, 173.95) | 50.56 |
| | | 0.2 | 1.37 | | |
| | | 0.1 | 9.80 | | |

[1] Corresponding extracts of all these species resulted in low cell viability (<30% SI) at the highest tested concentration (400 µg/ml)

[2] The major masses present in the bioactive fractions after microfractionation are shown within parentheses

[3] %SI of the bioactive fractions after microfractionation

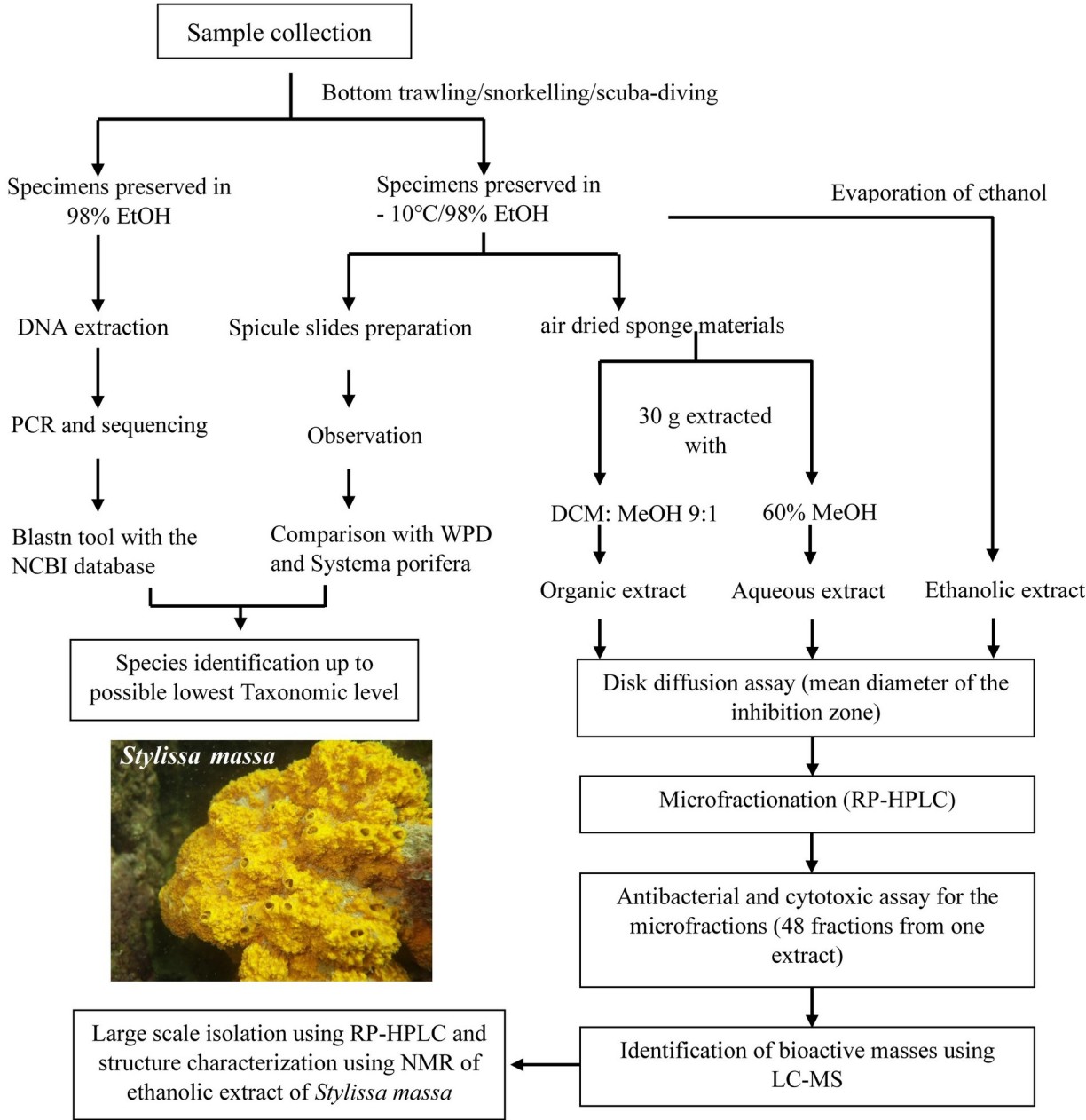

**Fig 2. Schematic diagram of the strategy used in marine sponge specimen identification and compound isolation from *Stylissa massa*.** Once the molecular ions associated with the antibacterial fractions were identified through the bioassay-guided microfractionation protocol, *Stylissa massa* was extracted in large scale, resulting in the isolation and structural characterization of bromopyrrole alkaloids. The underwater picture of *Stylissa massa* represents specimen 200220-01-11 which was used for large-scale extraction.

three known compounds: sceptrin, hymenin and either manzacidin A or C (stereoisomers indistinguishable by the acquired NMR data) (Fig 3).

All structures were confirmed by comparison of spectral data to previously published values [24, 27, 29]. After confirming the structures of isolated compounds as bromopyrrole alkaloids, sceptrin and hymenin were screened in the antibacterial and lymphoma cell toxicity assays. In the present study, sceptrin showed moderate antibacterial properties against both *E. coli* and *S.*

**Fig 3. Structures of bromopyrrole alkaloids isolated from *Stylissa massa*.** (**1**) Sceptrin; (**2**) Hymenin; (**3**) Manzacidin A/C. All structures were confirmed by comparison of spectral data to previously published values [24–29].

*aureus* (MIC = 62.5 μM) whereas hymenin exhibited a weak antibacterial activity (MIC = 250 μM) when compared with ciprofloxacin (MIC = 0.1–1 μM). Notably, no cytotoxicity was observed at 12.5–100 μM for compound sceptrin and hymenin against lymphoma cells.

## Discussion

In the present study, we investigated the chemodiversity and pharmacological potential of Sri Lankan marine sponges through bioassays for antibacterial and cytotoxic activities. To screen a large number of samples in an efficient manner, two methods were tested: the 51 crude extracts prepared were subjected to both (i) an agar disc diffusion assay and (ii) a microfractionation followed by microdilution assay. The agar disc diffusion assay identified seven antibacterial extracts, while the microdilution assay identified 21 microfractionated antibacterial extracts.

Of the 16 antibacterial species identified by the microdilution assay, antibacterial metabolites have so far been characterized from 11 species: *Erylus* sp., *Callyspongia (Cladochalina)* sp., *Aciculites orientalis*, *Topsentia* sp., *Acanthella* sp., *Petrosia (Petrosia)* sp., *Halichondria* sp., *Agelas ceylonica*, *X. testudinaria*, *Spheciospongia inconstans* and *S. massa* (S2 Table in S1 File). There are no records on characterization of any antibacterial metabolites from *A. gardineri*, *Amorphinopsis* sp. and *Tedania (Tedania)* sp., even though a few studies on antibacterial screening at crude level are found in the literature (S2 Table in S1 File). Aside from these, *A. ceylonica*, *A. orientalis*, *Manihinea* sp. and *R. indica* are also understudied species, with each seldomly investigated for natural products and only a few studies reporting characterized compounds (S2 Table in S1 File). One of the most potent antibacterial activities was shown by the organic extract of *A. orientalis* against *S. aureus* (with strong inhibition of bacteria even at a 32-fold dilution from the initial microfractionated well). The only compounds previously reported for *A. orientalis* are the antimicrobial peptides aciculitins A-C from its aqueous extract [30]. In the present work, microfractionated aqueous extract of *A. orientalis* was not antibacterial and aciculitin-type masses could not be observed in the aqueous extract. This suggests that latter compounds are either completely absent or existing in insignificant amounts. However, lack of information on any lipophilic compounds from this species highlights the importance of extending isolation attempts of its lipophilic extract. Similarly, *A. ceylonica*, which was identified to contain antibacterial compounds, is also an understudied species. Previously, the methyl ester of the alkaloid hanishin was isolated from a specimen collected in India and shown to be moderately antibacterial against *B. subtilis* [31]. This appears to be the

only other chemical study conducted on this species. The meroditerpenoid Chevalone E, is the only known antibacterial compound recorded from the genus *Rhabderemia*, which was isolated from the fungus *Aspergillus similanensis* KUFA 0013 associated with the sponge [32].

Although the extracts of *Manihinea* sp. was no longer cytotoxic following microfractionation (%SI = 59.16), it is still noteworthy of further exploration. Indeed, this species belongs to the Theonellidae family, from which numerous bioactive compounds (especially polyketides and macrolides) are described [33]. So far, only the cytotoxic tetramic acid glycoside aurantoside C is reported from *Manihinea lynbeazleyae* (Fromont & Pisera, 2011), one of the two known species in this genus [34]. *Acanthella*, which was highlighted to contain cytotoxic compounds in this study, is a well-studied genus with known cytotoxic secondary metabolites (S2 Table in S1 File) [35, 36]. In our sponge collection, *Acanthella* was a large specimen (S1 Fig in S1 File) that was trawled by the fish survey at 50 m depth, ~20 km off Wadduwa (West coast). A second specimen (not included in this study) was trawled on the East coast at 55 m depth, therefore this species seems to be present in relatively shallow waters around Sri Lanka and could be targeted for further studies.

The lack of cytotoxicity following microfractionation of some extracts may be explained by synergistic effects of compounds at the crude level. Alternatively, the potential bioactive compounds could be sensitive to fractionation and may have degraded during isolation. Although several promising species were identified from microfractionation, the lack of sufficient biomass for several species hampered further isolation work.

Guided by the parent molecular ions suggested from the bioactive microfractionated wells, three compounds were successfully isolated from *S. massa*: sceptrin, a well-known compound with a broad range of bioactivities, along with hymenin and manzacidin A/C. These three compounds are bromopyrrole alkaloids, all previously isolated from *S. massa* [37, 38]. Sceptrin is reported to have broad spectrum antimicrobial activity in a agar disc diffusion assay at 500 μg/disc [39]. We obtained moderate antibacterial activity for sceptrin (MIC = 62.5 μM) in the MHB microdilution assay. Similarly, hymenin also reported to have antibacterial properties against *B. subtilis* and *E. coli* with an inhibition zone diameter of 12 ± 0.3 and 13 ± 0.3 mm respectively at a concentration of 10 μg/disc besides its α-adrenoceptor blocking properties [25]. In the current work, hymenin gave only weak antibacterial activity in the MHB microdilution assay (MIC = 250 μM). This disparity in antibacterial activities across different assays could be due to variable assay conditions (type of media used, different incubation times, deviations in experimental handling of bacteria such as delay during transfer from culture to assay). Sceptrin isolated from *Agelas mauritiana* is reported to elicit its mechanism of action on the bacterial cell wall, with subsequent damage to the membrane. At the concentrations exceeding the MIC, sceptrin induced the formation of unusual spheroplasts in bacteria. The ability of sceptrin to affect the cell wall integrity of both prokaryotic and eukaryotic cells were confirmed by the release of potassium ions from *E. coli* and the lysis of defibrinated sheep red blood cells respectively [40]. Moreover, other bromopyrrole alkaloids such as axinellamine A and B (previously isolated from *Axinella* sp.) have shown broad spectrum antibacterial activity and an aberrant bacterial cellular morphology consistent with destabilization of bacterial secondary membrane and inhibition of normal septum formation [41]. No antibacterial activity was observed for manzacidin A/C in our hands, in line with what is reported in literature and no prominent bioactivity is reported for manzacidin A in previous reports, except weak protein kinase (VEGF-R2) inhibitor effect [28].

Of the two compounds, sceptrin is reported to have an inhibitory effect on cell motility of a variety of cancer cell lines [42] but no cytotoxicity effects are reported for hymenin. Notably, no cytotoxicity was observed up to 100 μM for both sceptrin and hymenin against lymphoma cells in our bioassay conditions. Although sceptrin has been subjected to a wide range of

bioassays, a few bioactivities are reported for hymenin and manzacidin A/C according to the literature. Thus, the latter two compounds should be assessed in a wider spectrum of bioassays. Fortunately, *S. massa* is a massive, common and widely distributed species in Sri Lanka, that can be easily collected by snorkelling, which makes it an easy target species for continuation of natural product research in the future.

Unfortunately, two major identified molecular ions (monoisotopic masses of *m/z* 324.2352 and 1037.8091) from *S. massa* could not be isolated, due to insufficient quantities. Some compounds, particularly those containing basic nitrogen atoms, are highly ionisable and might appear as abundant peaks even though they exist in minute quantities in the sponge material. With these factors considered, the microfractionation approach in combination with MS analysis of bioactive molecular peak ions is an effective tool to screen a large number of specimens to guide specialized metabolite isolation.

None of the reported bioactive molecular ion peaks in the literature from the same species (S2 Table in S1 File) corresponded with the monoisotopic masses of bioactive fractions identified herein, except *Stylissa massa* (Tables 2 and 3). One explanation is that several biotic and abiotic factors can likely affect the production of natural products, thus, different specimens from the same species collected in different locations may not show the exact metabolite profile. In fact, even the same sponge may not produce the same chemical constituents during its whole life span. Additionally, the target masses identified in the bioactive fractions may not necessarily be the exact parent monoisotopic mass of interest due to a combination of masses resulting from potential hydration, adduct formation and dimerization of the bioactive compound. This could result in a mismatch between the reported monoisotopic masses of bioactive metabolites from the same species or genus and the target masses present in the bioactive fractions. Consequently, this could complicate isolation attempts when a single bioactive mass is targeted during isolation. These factors need to be considered when using the potential target bioactive molecular ion peaks reported in Tables 2 and 3 as a guide for large-scale isolation.

## Conclusions

Altogether, this study highlights the chemodiversity and pharmaceutical potential of Sri Lankan marine sponges, advocating for future specimen collection and large-scale extraction. By using HPLC coupled with mass spectrometry and bioassays, the microfractionation protocol facilitated biologically interesting specimen selection for large scale analysis. Nine specimens (*Erylus* sp., *Callyspongia (Cladochalina)* sp., *Aciculites orientalis*, *Topsentia* sp., *Petrosia (Petrosia)* sp. 1, *Petrosia (Petrosia)* sp. 2, *Amorphinopsis* sp., *Acanthella* sp., and *Stylissa massa*) were identified to contain potent antibacterial activity, while one species (*Acanthella* sp.) stood out as the most promising species with high cytotoxic activity. Based on literature, seven species with antibacterial and/or cytotoxic activity (*Aulospongus gardineri*, *Amorphinopsis* sp., *Tedania (Tedania)* sp., *Agelas ceylonica*, *A. orientalis*, *Rhabderemia indica*, *Manihinea* sp.) were identified as understudied species for natural products, supporting the notion that Sri Lankan sponges offer an untapped source of new chemistry for drug discovery.

## Supporting information

**S1 File.** Includes images of the sponge species investigated in the current study, map of sponge sample collection sites in Sri Lanka, results of the agar disc diffusion assay, microfractionation exemplified by the isolation and identification of bromopyrrole alkaloids from *Stylissa massa*, a summary of sponge extracts that gave no antibacterial activity in the disc diffusion and microdilution assays and a literature review of antibacterial and cytotoxic compounds

identified in previous reports for the same sponge species as in the current study.
(DOCX)

**S1 Fig.**
(TIF)

## Acknowledgments

Marine sponge samples collected through the scientific surveys with the R/V Dr. Fridtjof Nansen as a part of the collaboration between the EAF-Nansen Programme and the Government of Sri Lanka were included in the study. The EAF Nansen Programme is a partnership between the Food and Agriculture Organization of the United Nations (FAO), the Norwegian Agency for Development Cooperation (Norad), the Institute of Marine Research (IMR), Norway, and partner countries and institutes e.g. National Aquatic Resources Research and Development Agency (NARA), Sri Lanka. We thank the EAF Nansen Programme and the Government of Sri Lanka for their immense support. We would like to thank the crew and scientists who participated in the *Dr. Fridtjof Nansen* Survey, and the National Aquatic Resources research and development Agency (NARA) for supporting the study in many ways especially, K. G. S. Nirbadha, Chaminda Karunarathne and Chaminda Prasad for assisting sponge sample collections.

## Author Contributions

**Conceptualization:** Chamari Hettiarachchi, Paco Cárdenas, Sunithi Gunasekera.

**Data curation:** Paco Cárdenas.

**Formal analysis:** Lakmini Kosgahakumbura, Jayani Gamage, Luke P. Robertson, Paco Cárdenas, Sunithi Gunasekera.

**Funding acquisition:** Chamari Hettiarachchi, Paco Cárdenas, Sunithi Gunasekera.

**Investigation:** Lakmini Kosgahakumbura, Jayani Gamage, Luke P. Robertson, Taj Muhammad, Chamari Hettiarachchi, Paco Cárdenas, Sunithi Gunasekera.

**Methodology:** Lakmini Kosgahakumbura, Jayani Gamage, Luke P. Robertson, Taj Muhammad, Björn Hellman, Ulf Göransson, Chamari Hettiarachchi, Paco Cárdenas, Sunithi Gunasekera.

**Project administration:** Chamari Hettiarachchi, Paco Cárdenas, Sunithi Gunasekera.

**Resources:** Ulf Göransson, Prabath Jayasinghe, Chamari Hettiarachchi, Paco Cárdenas, Sunithi Gunasekera.

**Supervision:** Taj Muhammad, Björn Hellman, Ulf Göransson, Chamari Hettiarachchi, Paco Cárdenas, Sunithi Gunasekera.

**Validation:** Lakmini Kosgahakumbura, Jayani Gamage, Chamari Hettiarachchi, Paco Cárdenas, Sunithi Gunasekera.

**Visualization:** Lakmini Kosgahakumbura, Jayani Gamage, Paco Cárdenas, Sunithi Gunasekera.

**Writing – original draft:** Lakmini Kosgahakumbura, Jayani Gamage, Luke P. Robertson, Paco Cárdenas, Sunithi Gunasekera.

**Writing – review & editing:** Lakmini Kosgahakumbura, Jayani Gamage, Luke P. Robertson, Taj Muhammad, Björn Hellman, Ulf Göransson, Prabath Jayasinghe, Chamari Hettiarach-chi, Paco Cárdenas, Sunithi Gunasekera.

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
