## [Decision Letter · Decision Letter 0]

12 Sep 2023

PONE-D-23-15586Screening for antibacterial and cytotoxic activities of Sri Lankan marine sponges through microfractionation: Isolation of bromopyrrole alkaloids from Stylissa massaPLOS ONE

Dear Dr. Sunithi Gunasekera,

Thank you for submitting your manuscript to PLOS ONE. After careful consideration, we feel that it has merit but does not fully meet PLOS ONE’s publication criteria as it currently stands. Therefore, we invite you to submit a revised version of the manuscript that addresses the points raised during the review process.

We look forward to receiving your revised manuscript.

Kind regards,

Awatif Abid Al-Judaibi, PhD

Academic Editor

PLOS ONE

Journal Requirements:

"This work was supported by Swedish Research Council's Linkage grant (2017-05416) awarded to S.G.  Marine sponge samples collected through the scientific surveys with the R/V Dr. Fridtjof Nansen as a part of the collaboration between the EAF-Nansen Programme and the Government of Sri Lanka were included in the study. The EAF Nansen Programme is a partnership between the Food and Agriculture Organization of the United Nations (FAO), the Norwegian Agency for Development Cooperation (Norad), the Institute of Marine Research (IMR), Norway, and partner countries and institutes e.g. National Aquatic Resources Research and Development Agency (NARA), Sri Lanka. We thank the EAF Nansen Programme and the Government of Sri Lanka for their immense support. We would like to thank the crew and scientists who participated in the Dr. Fridtjof Nansen Survey, and the National Aquatic Resources research and development Agency (NARA) for supporting the study in many ways especially, K. G.S. Nirbadha, Chaminda Karunarathne and Chaminda Prasad for assisting sponge sample collections. Additionally, this study made use of the NMR Uppsala infrastructure, which is funded by the Department of Chemistry–BMC and the Disciplinary Domain of Medicine and Pharmacy, Uppsala University."

"The research was funded by Swedish Research Council (https://www.vr.se/english.html) under an international collaborative linkage grant awarded to S. G. as the principal investigator, C.H. as the international partner and P.C. as the collaborating scientist (Award number:2017-05416). 

6. We note that Figures 2 and S1 in your submission contain copyrighted images. All PLOS content is published under the Creative Commons Attribution License (CC BY 4.0), which means that the manuscript, images, and Supporting Information files will be freely available online, and any third party is permitted to access, download, copy, distribute, and use these materials in any way, even commercially, with proper attribution. For more information, see our copyright guidelines: http://journals.plos.org/plosone/s/licenses-and-copyright.

a. You may seek permission from the original copyright holder of Figures 2 and S1 to publish the content specifically under the CC BY 4.0 license. 

7. We note that Figure S2 in your submission contain map image which may be copyrighted. All PLOS content is published under the Creative Commons Attribution License (CC BY 4.0), which means that the manuscript, images, and Supporting Information files will be freely available online, and any third party is permitted to access, download, copy, distribute, and use these materials in any way, even commercially, with proper attribution. For these reasons, we cannot publish previously copyrighted maps or satellite images created using proprietary data, such as Google software (Google Maps, Street View, and Earth). For more information, see our copyright guidelines: http://journals.plos.org/plosone/s/licenses-and-copyright.

a. You may seek permission from the original copyright holder of Figure S2 to publish the content specifically under the CC BY 4.0 license.  

Reviewers' comments:

Reviewer's Responses to Questions

**Comments to the Author**

1. Is the manuscript technically sound, and do the data support the conclusions?

Reviewer #1: Yes

Reviewer #2: Yes

2. Has the statistical analysis been performed appropriately and rigorously? 

Reviewer #1: N/A

Reviewer #2: Yes

3. Have the authors made all data underlying the findings in their manuscript fully available?

Reviewer #1: Yes

Reviewer #2: Yes

4. Is the manuscript presented in an intelligible fashion and written in standard English?

Reviewer #1: Yes

Reviewer #2: Yes

5. Review Comments to the Author

Reviewer #1: In this paper the authors have identified 35 sponge specimens from Sri Lanka and have identified 19 species to have cytotoxic and antibacterial activities. They have identified potent antibacterial and cytotoxic compounds from the aqueous extracts of Aciculites and Acanthella species. Extraction from Stylissa massa led to identification of bromopyrrole alkaloids with anti-bacterial activities. They have identified these bioactive compounds from less studied biodiversity hotspot in Sri lanka which could potentially have clinical use if characterized thoroughly.

Although they have identified several compounds with antibacterial and cytotoxic activity, the authors haven’t shown the mode of action of these drugs and their broad range activities against different bacteria or cell lines. Also the writing of the paper and the representative figures needs to be revisited and improved as it is hard for the readers to get the grasp of important results that are represented in table.

Major comments:

1. Results are not clearly represented for both antibacterial assay and cytotoxic assay. For table 2, Results of disk diffusion and microdilution assay would be best represented suing bar graphs as the numbers are difficult to interpret between species when represented in table.

2. Although the compounds are tested against 4 species only the results are shown for E. coli and S. aureus but not for Pseudomonas and Bacillus that were tested.

3. The final concentration of the compounds tested for diffusion assay is 20 ug/ml while the positive control ampicillin and gentamycin are used at the concentration of 10 ug/ml. Similar concentrations should be used for control and test compounds for comparison and dilutions of different concentrations must be tested.

4. For the compounds 1 and 2 tested why was the MIC was too high at 62.5 uM although it was previously shown to have antibacterial activity. Is it due to the mode of isolation from the extracts. Pure compounds (commercially synthesized) of 1 and 2 should be tested for anti-bacterial and cytotoxic effects to nullify any effect that has occurred during isolation.

5. As a follow up of previous point, though the mass and the structure has been determined by NMR, are the compounds tested with pure isolates of scepterin and hymenin to ascertain that both the isolated one from Stylissa massa and the pure compounds are same and runs similarly in HPLC?.

6. The authors have described the cytotoxic effects only in lymphoma cell lines. The effect of compounds on other cell lines including the non- carcinogenic ones like Hela must be tested as well to show that effect of tested compounds are specific to oncogenic cell line.

7. The mode of action of these antibacterial and cytotoxic compounds are not determined or discussed at all in the Discussion section. The effect of similar compounds on anti-bacterial and cytotoxic effects, the pathway it affects must be discussed.

8. Can the extracts from Aciculites and Acanthella species be repeated and scaled up as they had most antibacterial and cytotoxic activity?

Minor Comments

Line 27 – Change evaluated in to evaluated for

Line 32-33- Remove the sentence in parantheses as the details aren’t necessary for the abstract

What does (1) 2, 3 mean why it is parenthesized although the name has been described.

Line 39- Porifera –porifera

Line 564-565 – Why is Petrosia mentioned in paranthesis

Line 565- Remove paranthesis.

Table 1- What does voucher number, museum number and the Division Reg number mean in the table?

Table 1- what does the last column refer to what is COI? Is the numbers in the column refer to genbank accession number? How are the other species identified that doesn’t have any number eg: Row 1, Row 3 etc.

Table 2: It contains a lot of information. It would be good to represent them in graph for different assays or for different species.

.Line 302 and 304- Mention the actual final concentration instead of fold dilutions.

Reviewer #2: The systematic approach you employed in screening antibacterial and cytotoxic activities of marine sponges showcases the depth of your scientific rigor. The clarity and precision with which you detailed your methods and results enhance the reproducibility of your study.

Furthermore, your adeptness in isolating bromopyrrole alkaloids from Stylissa massa adds significant value to the field. Your findings contribute not only to the understanding of marine natural products but also to the potential development of novel therapeutic agents.

Congratulations on your exceptional work, and I eagerly anticipate seeing your research in print.

6. PLOS authors have the option to publish the peer review history of their article (what does this mean?). If published, this will include your full peer review and any attached files.

Reviewer #1: No

Reviewer #2: **Yes: **Lama Al_Darwish

---

## [Author Response · Author response to Decision Letter 0]

18 Oct 2023

Editor’s comments:

1- Please ensure that your manuscript meets PLOS ONE's style requirements: 

We have carefully revised our manuscript to ensure compliance with PLOS ONE’s style requirements. We believe our manuscript now aligns with the specified guidelines.

2- We suggest you thoroughly copyedit your manuscript for language usage, spelling, and grammar: 

We have carefully gone through the manuscript and carried out language editing. We confirm that we did not engage a professional scientific editing service, as we are satisfied with the language and clarity in our manuscript.

3- Please remove any funding-related text from the manuscript and let us know how you would like to update your Funding Statement. Currently, your Funding Statement reads as follow “The research was funded by Swedish Research Council (https://www.vr.se/english.html) under an international collaborative linkage grant awarded to S. G. as the principal investigator, C.H. as the international partner and P.C. as the collaborating scientist (Award number:2017-05416). The funders had no role in study design, data collection and analysis, decision to publish, or preparation of the manuscript”. 

We are content with the current funding statement. Additionally, we have removed any redundant funding information from other sections.

4- We note that you have stated that you will provide repository information for your data at acceptance. Should your manuscript be accepted for publication, we will hold it until you provide the relevant accession numbers or DOIs necessary to access your data. If you wish to make changes to your Data Availability statement, please describe these changes in your cover letter and we will update your Data Availability statement to reflect the information you provide.

All the relevant data are in the manuscript and supporting files. We apologize for this mistake for ticking the wrong box with respect to giving the accession numbers later. We have removed the ‘tick’ now.

Molecular markers (Folmer fragment of CO1 and/or 28S rRNA genes) were submitted to GenBank, accession numbers of these sequences are given in Table 1. 

5- Please include your full ethics statement in the ‘Methods’ section of your manuscript file. In your statement, please include the full name of the IRB or ethics committee who approved or waived your study, as well as whether or not you obtained informed written or verbal consent. If consent was waived for your study, please include this information in your statement as well: 

We would like to clarify that we did not conduct any tests on animal models requiring ethical clearance. Collection of sponges in their environment was carried out with written consent from the Wild life Department, Sri Lanka. We have added this information in the Material and Methods section. Our study then focused on screening sponge extracts for antibacterial and cytotoxic activities using microbe and human cell line assays, thus obviating the need for ethical clearance in our research.

6- We note that Figures 2 and S1 in your submission contain copyrighted images. All PLOS content is published under the Creative Commons Attribution License (CC BY 4.0), which means that the manuscript, images, and Supporting Information files will be freely available online, and any third party is permitted to access, download, copy, distribute, and use these materials in any way, even commercially, with proper attribution. We require you to either (1) present written permission from the copyright holder to publish these figures specifically under the CC BY 4.0 license, or (2) remove the figures from your submission: 

All the images presented in Figure 2 and S1 were captured by the authors of this manuscript. As a result, we assert that we do not require copyrights for these images.

7- We note that Figure S2 in your submission contain map image which may be copyrighted:

Figure S2 map was created by one of the authors of this manuscript using freely downloaded shapefiles of Grama Niladari Boundary Map of Sri Lanka, Survey Department, Sri Lanka and the final map generated using the freely available software GRASS GIS 7.6.1. Therefore, please note that there is no need to address copyright concerns for this figure.

Reviewer 1’s comments:

Major Comments:

1- Results are not clearly represented for both antibacterial assay and cytotoxic assay. For table 2, Results of disk diffusion and microdilution assay would be best represented using bar graphs as the numbers are difficult to interpret between species when represented in table:

We agree with the reviewer that the submitted Table 2 appeared complex, however it is difficult to present the data of microdilution assay in a bar graph, as suggested by the reviewer. Therefore, a bar graph is included in the supplementary section only to represent the data of disc diffusion assay. Furthermore, we have transformed the data in Table 2 into a simplified table enhancing the clarity and accessibility of our findings: the number of columns has been reduced, with the second column representing the extract type, and the next two columns representing the extract's activity in the disc diffusion assay and the microdilution assay, respectively. The titles of the two sub columns that represent the microdilution assay data was updated to make it easier for understanding. The previous table included the results of all the extract types of each bioactive specimen, regardless of whether they showed activity or not. The current table consists of only the bioactive extracts, making it more concise.

2- Although the compounds are tested against 4 species only the results are shown for E. coli and S. aureus but not for Pseudomonas and Bacillus that were tested:

We acknowledge that no activity was observed against Pseudomonas aeruginosa and Bacillus cereus, and we have included a sentence in the results section to report this lack of activity (line 285 and 286).

3- The final concentration of the compounds tested for diffusion assay is 20 µg/ml while the positive control ampicillin and gentamycin are used at the concentration of 10 µg/ml. Similar concentrations should be used for control and test compounds for comparison and dilutions of different concentrations must be tested:

 In the disc diffusion assay, the minimum concentration which showed the antibacterial activity for sponge extract is 20 ug/disc. Hence, we had to use all sponge extracts above 20 ug/disc for antibacterial activity. However, the control compounds were active at 10 ug/disc. We have included a sentence to clarify this in the manuscript. 

4- For the compounds 1 and 2 tested why was the MIC was too high at 62.5 uM although it was previously shown to have antibacterial activity. Is it due to the mode of isolation from the extracts. Pure compounds (commercially synthesized) of 1 and 2 should be tested for anti-bacterial and cytotoxic effects to nullify any effect that has occurred during isolation.

We cannot properly answer the first part of the question as there are too many unknown variables. Perhaps the previous authors used a more sensitive strain, which can often result in discrepancies in MIC values. In general, the differences in MIC values across different assays could be due to variable assay conditions (type of media used, different incubation times, deviations in experimental handling of bacteria such as delay during transfer from culture to assay), which we have explained in the discussion. Here, we cannot also vouch for the accuracy of the experimental conditions and assay carried out by the other research group. We do not think it is relevant to purchase the compounds as commercially synthesised products as we have performed full NMR analysis on these compounds at the endpoint of isolation. Whether they come from a commercially synthetic source or whether they are isolated from nature is immaterial, as their fundamental chemical structure is the same, and their purity has been independently assessed by us. All structural determination was done by full 2D NMR and in all cases comparison of 13C/1H NMR spectral data to the literature values. Moreover, these natural products are rare, expensive and not readily available for purchase to perform the experiments suggested by the reviewer. 

5- As a follow up of previous point, though the mass and the structure has been determined by NMR, are the compounds tested with pure isolates of scepterin and hymenin to ascertain that both the isolated one from Stylissa massa and the pure compounds are same and runs similarly in HPLC?

See the comment above. All structures have been structurally elucidated by full NMR and HRESIMS analysis and compared to literature data, and as such, this is unnecessary. Comparison of NMR spectral data to previous literature values in conjunction with independent 2D NMR analysis is the highest reasonable level of certainty that is required within natural products isolation papers. The purchase of such rare natural products is often impossible, and when it is, it is extremely expensive, and not within the budget of this project, which we believe is unnecessary given we have validated the compound structures with NMR. 

6- The authors have described the cytotoxic effects only in lymphoma cell lines. The effect of compounds on other cell lines including the non- carcinogenic ones like Hela must be tested as well to show that effect of tested compounds are specific to oncogenic cell line: 

We do not follow the reviewer’s comments that ‘non-carcinogenic ones like Hela cells’ as Hela cells are a typical cancer cell line. We have conducted cytotoxicity testing on microfractions and extracts as part of our screening process. Our future plans include isolating active compounds and conducting further tests against carcinogenic and non-carcinogenic cell lines to assess specificity.

7- The mode of action of these antibacterial and cytotoxic compounds are not determined or discussed at all in the Discussion section. The effect of similar compounds on anti-bacterial and cytotoxic effects, the pathway it affects must be discussed:

We believe that the reviewer is referring to the isolated pure compounds sceptrin, hymenin and manzacidin A/C, which are bromopyrrole alkaloids. Of these, only a few reports have investigated the mechanisms of action of sceptrin. We have incorporated information about the mode of action on sceptrin and bromopyrrole alkaloids in general into the discussion section (lines 503-516).

8- Can the extracts from Aciculites and Acanthella species be repeated and scaled up as they had most antibacterial and cytotoxic activity:

We appreciate the reviewer’s comments and this is part of our future plans for follow up studies. Due to sample limitation, we only conducted the two-step dilution assay on Aciculites orientalis in the current study. However, obtaining samples on a larger scale, isolating the compounds, and repeating assays on purified compounds are part of our future research plans.

Minor Comments:

Line 27 – Change “evaluated in”to “evaluated for”:

Corrected.

Lines 32-33 – Remove the sentence in parantheses as the details aren’t necessary for the abstract.

Corrected.

 What does (1) 2, 3 mean why it is parenthesized although the name has been described.

 In the previously submitted version, we mentioned the name of the compound the first time the compound was introduced with the number and then referred to the number of the compounds afterwards. We realized that this is not a requirement by PLOS One and in the resubmitted version we have removed the number of compounds and referred to them only by their name to improve clarity. 

Line 43 – Porifera –porifera

Taxonomic phyla names should be capitalized. Therefore, "Porifera" should be used instead of "porifera".

Lines 300, 301, 444, 560 – Why is Petrosia mentioned in paranthesis?

 "Petrosia" is mentioned as "Petrosia (Petrosia)" because the term "Petrosia" within parentheses represents the subgenus name, which should be enclosed within parentheses in accordance with scientific naming rules.

Table 1 – What does voucher number, museum number and the Division Reg number mean in the table? 

The Colombo National Museum provides two registration numbers for each voucher we submit: a museum number and a division number.

Table 1 – What does the last column refer to what is COI? Is the numbers in the column refer to genbank accession number? How are the other species identified that doesn’t have any number eg: Row 1, Row 3 etc.

The "CO1" column refers to the GenBank accession number, which will be available after this manuscript is published. Species without a CO1 accession number were identified only using morphological characters, using spicules and tissue slide preparations.

Table 2 – It contains a lot of information. It would be good to represent them in graph for different assays or for different species.

Corrected.

Lines 308, 310 – Mention the actual final concentration instead of fold dilutions.

Since we do not know the initial concentration of the extract in each well, we have mentioned only the fold dilution.

Additional Comments from Editorial office, sent on October 17th:

1. Please upload a Response to Reviewers letter which should include a point by point response to each of the points made by the Editor and / or Reviewers. (This should be uploaded as a 'Response to Reviewers' file type.) 

We have uploaded a ‘Response to Reviewers’ file as a separate file as well as copied the contents in the ‘response to reviewers’ file, under the relevant items in the submitted files. 

2. We note your current Data Availability statement is:

"Yes - all data are fully available without restriction;

All relevant data are within the manuscript and its Supporting Information files."

However, we also note you selected the box marked:

"Tick here if the URLs/accession numbers/DOIs will be available only after acceptance of the manuscript for publication so that we can ensure their inclusion before publication."

Please clarify whether your relevant data are in the manuscript and its Supporting Information files? Do you intend to provide the relevant data via public repository if the paper is accepted?

We apologize for this mistake for ticking the wrong box with respect to providing the URL of data at a later stage. All the relevant data are in the manuscript and supporting files. We have removed the ‘tick’ now.

---

## [Decision Letter · Decision Letter 1]

4 Dec 2023

PONE-D-23-15586R1Screening for antibacterial and cytotoxic activities of Sri Lankan marine sponges through microfractionation: Isolation of bromopyrrole alkaloids from Stylissa massaPLOS ONE

Dear Dr. Gunasekera,

Thank you for submitting your manuscript to PLOS ONE. After careful consideration, we feel that it has merit but does not fully meet PLOS ONE’s publication criteria as it currently stands. Therefore, we invite you to submit a revised version of the manuscript that addresses the points raised during the review process.

We look forward to receiving your revised manuscript.

Kind regards,

Awatif Abid Al-Judaibi, PhD

Academic Editor

PLOS ONE

Journal Requirements:

Reviewers' comments:

Reviewer's Responses to Questions

**Comments to the Author**

1. If the authors have adequately addressed your comments raised in a previous round of review and you feel that this manuscript is now acceptable for publication, you may indicate that here to bypass the “Comments to the Author” section, enter your conflict of interest statement in the “Confidential to Editor” section, and submit your "Accept" recommendation.

Reviewer #1: All comments have been addressed

Reviewer #3: All comments have been addressed

2. Is the manuscript technically sound, and do the data support the conclusions?

Reviewer #1: Yes

Reviewer #3: Yes

3. Has the statistical analysis been performed appropriately and rigorously? 

Reviewer #1: N/A

Reviewer #3: Yes

4. Have the authors made all data underlying the findings in their manuscript fully available?

Reviewer #1: Yes

Reviewer #3: Yes

5. Is the manuscript presented in an intelligible fashion and written in standard English?

Reviewer #1: Yes

Reviewer #3: Yes

6. Review Comments to the Author

Reviewer #1: The manuscript has been revised and all of the comments have been addressed.

The following points needs to be revised.

Line 408 and 423- Change “literature values” to “Previously published values”.

Line 456- Change A, orientalis to A. orientalis

Combine lines 513 to 516 to a single sentence.

Reviewer #3: Dear Authors

It is recommended to carry out future in vivo animal studies to confirm the in vitro enzyme assay

7. PLOS authors have the option to publish the peer review history of their article (what does this mean?). If published, this will include your full peer review and any attached files.

Reviewer #1: No

Reviewer #3: **Yes: **Mahmoud A. Al-Sha'er

---

## [Author Response · Author response to Decision Letter 1]

5 Dec 2023

Editor’s comments:

1- Please review your reference list to ensure that it is complete and correct. If you have cited papers that have been retracted, please include the rationale for doing so in the manuscript text, or remove these references and replace them with relevant current references. Any changes to the reference list should be mentioned in the rebuttal letter that accompanies your revised manuscript. If you need to cite a retracted article, indicate the article’s retracted status in the References list and also include a citation and full reference for the retraction notice: 

We have carefully reviewed the reference list and confirm that it is complete and correct. None of the cited papers have been retracted. No changes were made to the reference list during revising. 

Reviewer 1’s comments:

1- Line 408 and 423- Change “literature values” to “Previously published values”:

Corrected 

2- Line 457- Change A, orientalis to A. orientalis:

Corrected

3- Combine lines 514 to 517 to a single sentence:

Corrected 

Reviewer 3’s comments:

1- It is recommended to carry out future in vivo animal studies to confirm the in vitro enzyme assay:

We would appreciate your suggestion and look forward to conducting in vivo animal studies as a future plan.

---

## [Editor Report · Decision Letter 2]

13 Dec 2023

Screening for antibacterial and cytotoxic activities of Sri Lankan marine sponges through microfractionation: Isolation of bromopyrrole alkaloids from Stylissa massa

PONE-D-23-15586R2

Dear Dr. Sunithi Gunasekera,

We’re pleased to inform you that your manuscript has been judged scientifically suitable for publication and will be formally accepted for publication once it meets all outstanding technical requirements.

Kind regards,

Awatif Abid Al-Judaibi, PhD

Academic Editor

PLOS ONE

---

## [Editor Report · Acceptance letter]

29 Dec 2023

PONE-D-23-15586R2 

PLOS ONE

Dear Dr. Gunasekera, 

I'm pleased to inform you that your manuscript has been deemed suitable for publication in PLOS ONE. Congratulations! Your manuscript is now being handed over to our production team.

Kind regards, 

on behalf of

Professor Awatif Abid Al-Judaibi 

Academic Editor

PLOS ONE